# The Clinical Relevance of the EPH/Ephrin Signaling Pathway in Pediatric Solid and Hematologic Malignancies

**DOI:** 10.3390/ijms25073834

**Published:** 2024-03-29

**Authors:** Elena Chatzikalil, Ioanna E. Stergiou, Stavros P. Papadakos, Ippokratis Konstantinidis, Stamatios Theocharis

**Affiliations:** 1Division of Pediatric Hematology-Oncology, First Department of Pediatrics, School of Medicine, National and Kapodistrian University of Athens, 11527 Athens, Greece; ehatzikali@gmail.com; 2Department of Pathophysiology, School of Medicine, National and Kapodistrian University of Athens, 11527 Athens, Greece; stergiouioa@med.uoa.gr; 3First Department of Pathology, School of Medicine, National and Kapodistrian University of Athens, 11527 Athens, Greece; stpap@med.uoa.gr; 4Department of Internal Medicine, University of Connecticut, Farmington, CT 06030, USA; konstantinidis@uchc.edu

**Keywords:** EPH, ephrins, pediatric, hematology, oncology, therapeutic targeting

## Abstract

Pediatric neoplasms represent a complex group of malignancies that pose unique challenges in terms of diagnosis, treatment, and understanding of the underlying molecular pathogenetic mechanisms. Erythropoietin-producing hepatocellular receptors (EPHs), the largest family of receptor tyrosine kinases and their membrane-tethered ligands, ephrins, orchestrate short-distance cell–cell signaling and are intricately involved in cell-pattern morphogenesis and various developmental processes. Unraveling the role of the EPH/ephrin signaling pathway in the pathophysiology of pediatric neoplasms and its clinical implications can contribute to deciphering the intricate landscape of these malignancies. The bidirectional nature of the EPH/ephrin axis is underscored by emerging evidence revealing its capacity to drive tumorigenesis, fostering cell–cell communication within the tumor microenvironment. In the context of carcinogenesis, the EPH/ephrin signaling pathway prompts a reevaluation of treatment strategies, particularly in pediatric oncology, where the modest progress in survival rates and enduring treatment toxicity necessitate novel approaches. Molecularly targeted agents have emerged as promising alternatives, prompting a shift in focus. Through a nuanced understanding of the pathway’s intricacies, we aim to lay the groundwork for personalized diagnostic and therapeutic strategies, ultimately contributing to improved outcomes for young patients grappling with neoplastic challenges.

## 1. Introduction

### 1.1. Challenges and Opportunities in Pediatric Neoplasms: A Comprehensive Overview

Pediatric neoplastic disease stands as the second leading cause of death and a prominent contributor to treatment-related morbidity in childhood [1]. The predominant cancer type in this demographic is leukemia, representing nearly 24% of all pediatric cancer cases [2], closely followed by Central Nervous System (CNS) neoplasms and lymphomas, accounting for 15% and 16% of cases, respectively [2,3]. Epithelial neoplasms and melanomas collectively contribute to 12% of all pediatric tumors, while hematologic malignancies (other than leukemia and lymphoma), rhabdomyosarcoma, thyroid carcinoma, and germ cell tumors each account for 6% [1]. Neuroblastoma, kidney tumors (including Wilms’ tumor), and liver tumors (including hepatoblastoma) represent 4%, 3%, and 1%, respectively [1,4].

Despite significant improvements in survivorship for certain neoplasms, like acute lymphoblastic leukemia (ALL) and Hodgkin lymphoma, not much progress has been made for other neoplasms, such as diffuse intrinsic pontine glioma (DIPG) and neuroblastoma, even after years of applying the available cancer therapies [1]. Our inability to reliably predict the clinical behavior of these neoplasms based on tumor histology poses a significant obstacle to the development of effective treatments [5]. Furthermore, the rarity of pediatric oncologic diagnoses and the exclusion of the pediatric population from clinical trials involving targeted therapy hinder the accumulation of clinical evidence regarding the efficacy of novel pharmaceutical agents against pediatric neoplastic disease [6].

Considering these challenges, there is a need to review the existing evidence on available targeted therapies in terms of safety and efficacy and, furthermore, evaluate novel targeted therapeutic options. This approach could be important in expanding the treatment options for pediatric neoplasms, particularly those with an unfavorable prognosis with current available therapies. The quest for improved understanding and targeted interventions is critical in advancing the care and outcomes for young patients grappling with neoplastic challenges.

### 1.2. Insights into EPH/Ephrin Cellular Signaling

In 1987, the molecular cloning of a novel kinase receptor gene named “*EPH*” (erythropoietin-producing hepatocellular receptor gene) in a human carcinoma cell line marked a pivotal discovery [7]. Currently, the erythropoietin-producing hepatocellular receptor (EPH) family stands as the largest and most expanding group of receptor tyrosine kinases (RTKs), characterized by a prototypical structure encompassing extracellular ligand-binding regions, hydrophobic transmembrane helices, and intracellular cytoplasmic domains hosting protein tyrosine kinase domains and various interaction sites [7,8]. The juxtamembrane region serves as a vital connection between the extracellular and intracellular domains, linked to a sterile alpha domain (SAM) and a post-synaptic density protein (PSD95), Drosophila disc large tumor suppressor (Dlg1), and zonula occludens-1 protein (PDZ)-binding domain [9].

EPHs are categorized into two subgroups, EPHAs [1,7,8,9,10,11,12,13,14] and EPHBs [1,7,8,9,13], totaling nine and five members, respectively, in humans. This categorization is based on sequence homology, structural features, and ligand-binding specificities [8,9]. Ephrin ligands are divided into ephrin-A and ephrin-B subgroups, encoded by *EFNA* and *EFNB* genes, respectively [8]. While EPHs adhere to the prototypical RTK features, ephrins exhibit distinct structures. EPHAs predominantly interact with ephrin-As, usually attached to the plasma membrane, and EPHBs preferentially interact with ephrin-Bs, linked to an intracellular PDZ-binding domain, featuring a transmembrane segment and a short cytoplasmic domain-tail [8].

Except for this characteristic interaction pattern described above, there are cases of cross-class binding in which EPHs (e.g., EPHA4, EPHB2) bind to ephrins of a different class [8,9,10,11,12,13,14,15,16,17,18,19,20,21]. Additionally, EPHA10 and EPHB6 exhibit modifications in conserved regions of their kinase domains that hinder kinase activity. Furthermore, various alternatively spliced forms identified for many EPHs deviate from the prototypical structure and have distinctive functions [22,23].

Though it has initially been believed that the EPH/ephrin signaling pathway is activated solely in close cell–cell interactions, deviations from this notion have been described. Despite the strong affinity between EPHs and ephrins during close cell–cell interactions, studies in cancer have demonstrated that these interactions are not consistently required. For instance, EPHA2 expression on extracellular vesicles can influence nearby ephrin-A1-expressing tumor cells [24]. Ephrin ligation to EPH triggers signaling in the cytoplasmic side. In the absence of the ligand interaction (“off” state), unphosphorylated tyrosine molecules near the cell membrane interact with the kinase domain, disrupting its structure and dysregulating tyrosine kinase activity. Conversely, in the presence of the ligand interaction (“on” state), EPHs undergo phosphorylation in the juxtamembrane tyrosine regions, blocking their interaction with the kinase domain and relieving its distortion. Consequently, downstream substrates undergo phosphorylation, initiating various signaling pathway cascades [25].

A distinct characteristic of EPH/ephrin complexes lies in their ability to generate bidirectional signals, affecting both the receptor-expressing and ephrin-expressing cells [7,8,9,25,26]. “Forward signaling” into the EPH-expressing cell relies on their tyrosine kinase domain, mediating autophosphorylation and phosphorylation of other proteins, and on the associations of the receptor with various effector proteins [22]. Simultaneously, “reverse signaling” into the ephrin-expressing cell depends in part on the tyrosine phosphorylation of the cytoplasmic region and on associated proteins [26].

Bidirectional EPH/ephrin signaling, along with processing modalities, like those mediated by metalloproteases (MMPs) followed by γ-secretase-mediated cleavage, contributes to various physiological and tumorigenic processes [27]. Furthermore, it is noteworthy that EPHs and ephrins can signal independently or crosstalk with other signaling systems, producing diverse outcomes in human cellular signaling networks [28]. Characteristic examples of synergistic function with other cellular signaling systems are epidermal growth factor (EGF) and fibroblast growth factor (FGF) crosstalk with EphA2 and EphA4, respectively, communication between EPHB/ephrin-B members and Wnt signaling, and the EPH/ephrin interaction with E-cadherin and integrin, tight junctions, and calcium channels, all of which regulate the EPH/ephrin expression or cells’ motility and proliferation capacity, most of the times independently of EPH/ephrin system modulation. Despite the variety in their synergistic functions, bidirectional signaling is their major and best-characterized form of communication and functionality [29,30].

The structure and interactions of the different EPHs and ephrins are illustrated in Figure 1.

## 2. EPH/Ephrin Signaling in Physiologic Processes and Tumorigenesis

The EPH/ephrin signaling pathway, now comprising nearly 25% of the 58 human RTKs [31], orchestrates a myriad of physiologic and homeostatic events, contributing significantly to tissue organization in higher organisms. Found in most tissues in a combinatorial manner with dynamically changing expressions [32], the EPH/ephrin system plays a pivotal role in neurogenesis and neuronal migration during embryogenesis, guiding axonal growth in early brain development [33]. Evidence suggests that ephrin-B3 expression inhibits cell cycle progression and apoptosis in the adult subventricular zone, impacting adult neurogenesis [34]. Negative cell cycle regulation is also observed in various cell types, including thymic epithelial cells, bone marrow cells, as well as breast, gastrointestinal, and skin epithelial cells [35,36,37]. Additionally, the interaction between EPHB4 and ephrin-B2 is strongly implicated in angiogenesis and lymphangiogenesis [1], while EPH/ephrin signaling contributes significantly to the morphogenesis and differentiation of cells in certain tissues [38]. Remarkably, EPH-triggered responses induce cytoskeletal rearrangements, including cytoskeleton collapse, with forward signaling often resulting in cell repulsion and ephrin reverse signaling eliciting either cell repulsion or adhesion [33,39].

Turning to the tumorigenic landscape, multiple EPHs and ephrins are expressed both in cancer cells and the tumor microenvironment, facilitating aberrant cell–cell communication within and between tumor compartments [40,41,42,43]. Dysregulation of EPH/ephrin signaling in humans leads to congenital diseases and cancer [32]. Notably, EPHA2 upregulation is associated with an increased risk for malignancy and a poor clinical prognosis in various cancer types [42,43,44,45,46,47], and EPHB4 expression is widely observed in neoplastic cells, correlating with tumor progression [48,49,50]. The EPH/ephrin system intricately participates in tumorigenic processes, instigating metastatic development, angiogenesis, de-differentiation, cancer cell formation, and stem cell proliferation [40,51]. Of particular interest is its role in peripheral nerve invasion through its impact on the nerve microenvironment [52]. Furthermore, the EPH/ephrin signaling influences vessel infiltration, and its crucial role in epithelial–mesenchymal transition (EMT) has been well-established [26,52]. Considering this evidence, several studies have been conducted during the last decade, and especially during the last five years, including large single-cell analysis studies on the EPH/ephrin system, which collected and integrated gene expression data. Characteristic examples are Ravasio et al.’s single-cell analysis, which investigated the role of activated EPHA receptors in multiple carcinoma cells, providing a novel scoring system for EPHA clustering that can assay the heterogeneity of tumor cell types, and Shen et al.’s single-cell and transcriptomics study, which revealed EFNA1 as a potential prognostic marker for cervical cancer [53,54]. 

Acquiring a detailed understanding of how cells interact, both in their usual and problematic states through the EPH/ephrin axis, holds promise for developing novel approaches to enhance pediatric cancer treatment.

## 3. Exploring EPH/Ephrin Signaling in Solid Tumors of Pediatric Patients

There is a growing body of evidence supporting the pivotal role exerted by the EPH/ephrin signaling pathway in the progression of solid tumors, facilitating cell migration, proliferation, and invasive properties [51,55]. Solid tumors, encompassing pediatric solid neoplasms, involve the infiltration of normal tissue by tumor cells with invasive potential, complicating therapeutic decisions and promoting metastatic development [56]. The predominant feature of pediatric solid tumors is the local recurrence attributed to tumor cell migration and invasion. Understanding the specific involvement of the EPH/ephrin signaling pathway in the development of the most prevalent pediatric tumors could prove beneficial in shaping new therapeutic protocols. Some members of the EPH/ephrin family, identified as upregulated in various types of pediatric solid tumors, present promising prospects as therapeutic targets [51,52,55]. Nevertheless, the comprehensive exploration of EPH/ephrin members’ expression in pediatric solid neoplasms remains limited.

Our literature review explores the findings related to EPH/ephrin expression and its impact on the most common pediatric solid tumors.

### 3.1. Brain Tumors

The EPH/ephrin signaling pathway plays a pivotal role in both the embryonic developmental function of the human brain and the homeostatic function of the adult human brain. During the development of the embryonic nervous system, the bidirectional signaling of EPH/ephrin members (EphA4, EphB1, EphB2, EphB3, and ephrin-B1) aids in interneuron migration, progenitor cell adhesion, dendritic spine formation, and synaptic maturation [57,58,59]. In adulthood, various EPH/ephrin members regulate either positively (EphA4, EphA7) or negatively (EphA2, EphB1, EphB3) CNS angiogenesis, neuron migration, and proliferation to maintain homeostasis [60,61]. Moreover, EphA7, EphB1, EphB2, ephrin-A2, ephrin-A3, and ephrin-A5 are involved in neurogenesis processes in the adult life, particularly in the subgranular zone [62,63,64]. Disruption of these mechanisms can lead to a disturbance in homeostasis, resulting in various pathological conditions, including tumorigenesis, due to an ineffective balance of angiogenetic and neurogenetic events. This section examines the tumorigenic activity of EPH/ephrin members in various types of pediatric brain tumors.

#### 3.1.1. Medulloblastoma

Medulloblastoma is the most common malignant brain tumor in childhood. It is currently treated with maximal resection, chemotherapeutic agents, with or without autologous stem cell transplantation in ongoing multicenter clinical trials, and craniospinal irradiation [65]. Ongoing research focuses on biological-based risk stratification as a result of a better understanding of the tumor’s molecular characteristics. 

Regarding EPH/ephrin members’ expression, upregulation of EPHB2 and ephrin-B1 has been reported in tumor samples compared to normal cerebellum, while EPHA2, EPHB2, and EPHB4 showed overexpression in medulloblastoma cell lines [66,67]. Moreover, subsequent studies have confirmed the crucial role of EPH/ephrin signaling in the medulloblastoma metastatic capacity; EPHB2 and ephrin-B1—except for being overexpressed—were also found to associate with invasive and metastatic tumor subtypes [67]. Mechanistically, ephrin-B1 induced the phosphorylation of EPHB2 and, to a lesser extent, EPHB4, with p38, extracellular-signal-regulated kinase (ERK), and mammalian target of rapamycin (mTOR) being identified as downstream signaling mediators of EPHB2 activation [67]. Ephrin-B2 has been reported to be expressed in all medulloblastoma tumors, while ephrin-B1, associated with more aggressive phenotypes, was expressed in 20% of the tumors [68]. Further characterization by McKinney et al. identified a tumor-specific expression pattern for EPHB1, EPHB2, and ephrin-B1 in medulloblastoma. Considering that ephrin-B1 overexpression stimulates EphB activation, resulting in an alteration of F-actin distribution and morphology, attenuated adhesion, and enhancement of proliferation, the authors suggested ephrin-B1 as a potential target for the treatment of aggressive medulloblastoma subtypes resistant to conventional therapies [69]. Finally, studies using mouse models and medulloblastoma cells in culture to reveal the impact of EPH/ephrin signaling on the medulloblastoma invasive and metastatic capacity proposed ephrin-A5, which is involved in the phosphoinositide 3-kinase (PI3K)/Akt oncogenic pathway, and EPHB1, whose knockdown enhanced cellular radiosensitization, as promising therapeutic targets [70,71]. The EPH/ephrin-related molecular pathways implicated in the pathogenesis of medulloblastoma are illustrated in Figure 2A.

#### 3.1.2. Glioma

EPH/ephrin aberrations have been associated with glioma histological grade and tumorigenic processes [72]. Some members, such as EPHA2 and EPHA7, are overexpressed in glioma stem cells, indicating a poor prognosis [73,74,75]. Liu et al. demonstrated that ligand-mediated EPHA2 activation enhanced glioblastoma proliferation and tumor growth via a mitogen-activated protein kinase (MAPK) dependent pathway [74]. EPHA2 overexpression has also been shown to promote glioma cell migration in a ligand-independent manner, requiring its phosphorylation at serine 897 by Akt [76]. Experiments on human glioma U251 cells showed that EPHA4 promoted fibroblast growth factor 2 (FGF2)-mediated cell proliferation and migration, along with increased MAPK and Akt phosphorylation stimulated by FGF2. Additionally, EPHA4 overexpressing cells were characterized by an increase in the active forms of Rac1 and cell division control protein 42 homolog (Cdc42), while EPHA4 formed a heteroreceptor complex with fibroblast growth factor receptor 1 (FGFR1), inducing FGFR1-mediated downstream signaling [77]. Nakada et al. reported that EPHB2 phosphorylation promotes glioma cell growth, migration, and invasion via R-Ras mediated signaling [78,79]. Ephrin-B2 overexpression characterized glioma tissue specimens, while its phosphorylation was shown to promote tumor invasion, as well as angiogenesis via vascular endothelial growth factor receptor 2 (VEGFR2) regulation [80,81]. Ephrin-B3 has also been reported to characterize a glioma-aggressive phenotype, promoting invasion via Rac1 activation [82,83]. Conversely, overexpression of EPHB1 and ephrin-A2 is associated with a favorable prognosis [72], while ephrin-A5 acts as a tumor suppressor via the negative regulation of epidermal growth factor (EGFR) [84]. Various studies identified EPH members with either inhibitory (EPHB1, EPHB3) or promoting (EPHA7, EPHB2) effects on cell migration, proliferation, and invasion [85]. Of note, some EPHs, such as EPHA4, can exert both tumor-promoting and tumor-suppressive functions [77,86]. Regarding the Eph/ephrin signaling pathway exploitation in glioma therapeutic strategies, EPHA2 expression in high-grade gliomas makes it a potential therapeutic target, with compounds like Dasatinib showing promising results in patient survival [87,88]. Towards this direction, EPHB4/ephrin-B2 signaling is explored as an antiangiogenic target, with anti-EPHB4 monoclonal antibodies successfully blocking EPHB4 signaling [89,90,91], while inhibition of EPHA2 and EPHB1-3 forward signaling in gliomas is associated with a dose-dependent reduction in cell invasion [92]. The distinctive biological features of the EPH/ephrin signaling pathway present potential therapeutic targets for pediatric glioma treatments, especially those for high-grade tumors, which need more targeted approaches due to their biologic heterogeneity and poor prognosis with the current standard therapies (surgery, irradiation, and chemotherapy with temozolomide, either combined or not with targeted therapies in the field of pediatric clinical trials) [93]. Figure 2B summarizes the molecular mechanisms by which EPH/ephrin members exert their role in the development and progression of glioma. 

#### 3.1.3. Ependymoma

Pediatric ependymoma is a tumor subtype for which the standard treatment consists of gross total resection and adjuvant radiation therapy, while there is a lack of efficient chemotherapeutic agents [94]. The study of EPH/ephrin expression in ependymoma tumors is limited. In 2005, research identified the overexpression of ephrin-A3, ephrin-A4, EPHB2, EPHB3, and EPHB4 in ependymoma tumor cells, either from frozen or formalin-fixed tumor samples, but the clinical impact remains unclear [95]. Recent studies confirmed *EPHB2* as an ependymoma oncogene, suggesting that the dysregulation of receptor expression may contribute to tumorigenesis. EPH/ephrin signaling pathway activation (especially EPHB1-EPHB3) is higher during CNS maturation and decreases in adulthood, potentially playing a role in ependymoma development [96].

#### 3.1.4. Others

Except for the most common pediatric brain tumors, few studies report EPH/ephrin aberrations and targeting in pediatric patients’ groups diagnosed with some rarer neoplasms. For example, Gump et al. reported that EPHA2 overexpression in craniopharyngioma tumor cells, compared to normal cells, provided evidence for using EPHA2 as a therapeutic target [97]. Moreover, EPHA2 and EPHB1 are overexpressed in neurofibromatosis type 2 gene (*NF2*) deficient meningiomas, and their pharmaceutical targeting with Dasatinib favors patients’ prognosis [98]. 

Irrefutably, there is an urgent need for designing and conducting randomized clinical trials using targeted agents to treat rarer, as well as the most common, pediatric brain tumors in the near future. The above-described implications of the EPH/ephrin signaling pathway in the pathogenesis of pediatric brain tumors, as well as its potential therapeutic targeting options, are summarized in Table 1. 

### 3.2. Neuroendocrine Tumors

Research data underscore the pivotal role of EPH/ephrin signaling during the initial stages of peripheral nerve repair after transection [99]. EPH/ephrin members, and especially EphB2, guide Schwann cells in collective migration, forming multicellular cords that facilitate axon guidance across the injury site [99]. Errors in the mechanisms of cell migration and proliferation in the peripheral nervous system may potentially lead to tumorigenesis. Neuroblastomas stand as the most common tumors of the peripheral nervous system during childhood, contributing to approximately 15% of childhood cancer-related mortality [1]. These tumors display genetic, morphological, and clinical heterogeneity, posing challenges to the effectiveness of current treatment modalities, consisting of surgical resection, combination chemotherapy with or without stem cell transplantation, and radiation therapy [100]. Considering the above, the necessity for the establishment of novel therapeutic strategies is highlighted.

The role of EPH/ephrin members in neuroblastoma tumors was initially clarified by the research of Tang et al. [101,102]. In their first study, they identified the expression of EPHB6, ephrin-B2, and ephrin-B3 in neuroblastoma cells, correlating with a favorable prognosis [101]. In a subsequent study, *EPHB2* expression was demonstrated in low-stage neuroblastoma tumors, correlating with *MYCN* expression in tumors without *MYCN* amplifications [102]. Gene expression analysis via real time PCR (RT-PCR) conducted a few years later revealed variations in the *EPHA2* expression levels among different neuroblastoma cell lines [103]. Significantly, *EPHA2* expression decreased after pharmaceutical treatment with doxorubicin, indicating its potential as a biomarker for drug responsiveness in neuroblastoma [103]. Additionally, EPHB6 and EPHB1 expression analyses underscored their crucial role in tumorigenic activity [104,105]. Specifically, Chen et al. demonstrated that EPHB1 was post-translationally modified by the small ubiquitin-like modifier (SUMO) protein at lysine residue 785, with the SUMOylation resulting in the inhibition of EPHB1 downstream signaling via protein kinase C gamma (PKCγ), and subsequently in the repression of tumor growth [105]. EPHB4 was identified as a tumorigenic factor, with *EPHB4* gain correlating with advanced-stage neuroblastoma and poor overall survival (OS) [106]. Furthermore, the EPHB4-V871I variant contributed to increased proliferation, migration, and invasion properties in neuroblastoma cell lines via increased phosphorylation of the ERK1-2 pathway and effects on the vascular endothelial growth factor (VEGF), c-RAF, and cyclin-dependent kinase 4 (CDK4) target genes, while treatment with EPHB4 inhibitors reversed the phenotype driven by the variant, suggesting an option for therapeutic targeting in neuroblastoma [106]. Multi-targeted tyrosine kinase inhibitors, such as Dasatinib used in clinical trials, hold the potential to inhibit the kinase activity of EPHB4. Further analyses of expression are imperative to establish the role of the EPH/ephrin signaling pathway in neuroblastoma tumorigenesis and treatment, providing insights into new therapeutic strategies. The studies investigating the implication of EPH/ephrin members in neuroblastoma tumorigenesis are summarized in Table 2, while the molecular mechanisms mediating their functions are illustrated in Figure 3. 

### 3.3. Sarcomas

The EPH/ephrin signaling pathway is implicated in the growth process of different bone and soft tissue types, while it also controls angiogenetic processes during embryogenesis. Additionally, it governs mesenchymal stem cell migration, angiogenesis, and bone remodeling in adult life. Deregulation of these homeostatic functions due to the pathological expression of EPH/ephrins may contribute to tumorigenesis [107,108,109,110]. Pediatric sarcomas are treated with combined chemotherapy, surgery, and/or radiotherapy, but despite the patients’ improved prognosis in the last two decades, some subtypes remain incurable. Current studies are focusing on signaling pathways involved in sarcomas’ tumorigenesis to identify novel agents with the potential to overcome multidrug resistance [111].

#### 3.3.1. Ewing Sarcoma

Ewing sarcoma, a highly aggressive pediatric tumor affecting bones and soft tissues, relies on a functional vascular network for nutrient and oxygen delivery, as well as waste removal. Given the known roles of EPH/ephrins, particularly EPHA2, in promoting angiogenesis and tumorigenesis, their involvement in Ewing sarcoma pathogenesis, prognosis, and therapy is under exploration. The study by Sáinz-Jaspeado et al. in 2013 demonstrated that EPHA2 expression promotes endothelial cell migration and enhances the tumor’s angiogenetic activity. The researchers showed that the interaction between EPHA2 and caveolin-1 (CAV1) was crucial for the right localization and signaling of the receptor to induce AKT-mediated basic fibroblast growth factor (bFGF) production and subsequently promote endothelial cell migration [112,113]. Recent studies further establish the tumorigenic function of EPHA2 in Ewing sarcoma, with EphA2 ligand-independent activity controlled upon phosphorylation at S897, being linked to enhanced proliferation and migration capacities of the tumor cells [86]. Additionally, EPHA2 silencing dampened tumorigenicity, migration, and invasion in vitro, as well as the incidence of lung metastasis in experimental and spontaneous metastasis assays in vivo [86]. These findings highlight the significance of EPHA2-expressing cells in Ewing sarcoma propagation, unraveling the potential of EPHA2 targeting in anti-metastatic therapy [114]. Figure 4A illustrates the molecular pathways that have been shown to correlate with EPH/ephrin signaling in the pathogenesis of Ewing sarcoma. 

#### 3.3.2. Osteosarcoma

The molecular mechanisms driving osteosarcoma development remain elusive. The study of Fritsche-Guenther et al. revealed the upregulation of ephrin-A1 and the de novo expression of EPHA2 in osteosarcoma tissues [113]. Ephrin-A1 binding to EPHA2, which is not expressed in normal bone but is overexpressed in osteosarcoma, induces increased tyrosine phosphorylation and activation of downstream signaling pathways mediated by MAPK activation, possibly stimulating osteosarcoma’s metastatic potential [113]. EPHA2 knockdown has been shown to drive the inhibition of cell viability and migration in other types of sarcoma [112]. Ongoing clinical trials explore potential pharmaceutical targeting using selective multi-targeted tyrosine kinase inhibitors, such as pazopanib, for pediatric osteosarcomas [114,115,116]. Of note, Chiabotto et al. demonstrated that the combination of pazopanib and trametinib downregulated EPHA2 in osteosarcoma mouse models [117]. Additionally, EPHA7 regulation by microRNAs (miRs) has been shown to be involved in osteosarcoma pathogenesis, with EPHA7 exerting a tumor-promoting role. Specifically, EPHA7 was identified as the target of miR-101. Tu et al. demonstrated that the downregulation of human histocompatibility leukocyte antigen (HLA) complex P5 (HCP5), which targets miR-101, inhibits osteosarcoma cell proliferation, migration, and invasion via the competitive binding of miR-101 to EPHA7 [118]. More recently, Wu et al. reported that miR-488 suppresses osteosarcoma cell proliferation and invasion through EPHA7 targeting [119,120]. Finally, ephrin-A4 expression serves as a potential marker for identifying high-risk patients with a poor prognosis and inferior chemotherapy response [120]. Further studies and clinical trials are essential to unravel the potential of EPHs and ephrins as therapeutic targets for osteosarcoma. Figure 4B summarizes schematically the molecular mechanisms via which EPH/ephrin members participate in the tumorigenesis of osteosarcoma. 

#### 3.3.3. Rhabdomyosarcoma

Rhabdomyosarcoma, the most common soft tissue sarcoma in children, is often diagnosed with concurrent metastases, posing challenges for effective therapies. An increased expression of EPHs (EPHB1, EPHB2, EPHB3, and EPHB4) and their ephrin ligands (ephrin-B1 and ephrin-B2) has been implicated in promoting angiogenesis and tumor progression in rhabdomyosarcoma [121,122,123,124]. EPHB4, specifically, has been highlighted as a potential therapeutic target for alveolar rhabdomyosarcoma, with studies suggesting its role as a poor prognostic indicator in certain cell lines (especially A203, RD-ES, and A673) [122,123,124]. Ongoing research and clinical trials are crucial for confirming these findings and exploring the therapeutic potential of EPH/ephrin in rhabdomyosarcoma. 

Detailed insights into the role of EPH/ephrin signaling in sarcomas, particularly Ewing sarcoma, osteosarcoma, and rhabdomyosarcoma, along with potential therapeutic implications, are summarized in Table 3.

### 3.4. Renal Tumors (Wilms)

Various Eph/ephrin members are expressed in different parts of the normal renal system, each with distinct functions. Notably, EPHB2 is implicated in the genesis, differentiation, and developmental processes of renal system structures [125]. The understanding of parameters and biomarkers related to Wilms tumors’ metastatic potential and prognosis remains limited, while the current therapeutic options, using either upfront surgery or upfront chemotherapy with delayed surgery, present no difference in survival [126]. Specifically addressing the EPH/ephrin signaling pathway, Chetcuti et al. explored the expression profile of EPHB2 in Wilms tumor samples with RT-PCR and immunohistochemical (IHC) analysis [127]. The findings indicated an elevated expression of EPHB2 in invasive Wilms tumors stages 2–4 compared to stage 1, suggesting that EPHB2 could serve as a marker for the invasive capacity of Wilms tumors [127]. However, further investigations in larger cohorts are necessary to validate these observations. The details and insights pertaining to EPH/ephrins signaling in Wilms tumors are encapsulated in Table 4.

## 4. EPH/Ephrin Signaling in Hematologic Malignancies of Pediatric Patients

The pioneering identification of EPHB4’s involvement in hematopoietic processes, specifically myeloid differentiation, marked the initiation of understanding the implication of EPH/ephrin signaling in the context of hematopoiesis [128]. Subsequent findings have reported its role in erythroid and monocyte differentiation, as well as B-cell maturation [129]. However, the exploration of EPH/ephrin signaling in the oncogenesis of the pediatric hematopoietic system remains limited, with existing data predominantly focusing on leukemia [129]. In this section, we will outline the reported role of EPH/ephrins in pediatric hematologic malignancies, primarily focusing on T-cell ALL (T-ALL), which presents relapse in most pediatric patients, despite the use of chemotherapeutic agents, novel agents targeting lymphocyte signaling pathways (i.e., Notch1, JAK, and BCL2 inhibitors), steroids, and allogeneic stem cell transplantation [130].

In a study by Li et al., including 40 pediatric patients with ALL and 10 controls, protein expression analysis in bone marrow samples revealed a higher frequency of *EPHB4* methylation in ALL patients. This methylation profile negatively correlated with patients’ prognoses, therefore highlighting a tumor-suppressive effect of *EPHB4* [131]. Another study by El Zawily et al., comparing 117 pediatric T-ALL samples with the controls, reported an elevated EPHB6 expression in T-ALL patients. This increased expression was associated with an enhanced sensitivity to doxorubicin, resulting in an augmented apoptotic response via the preservation of Akt signaling [132]. Colluci et al. also confirmed a higher EPHB6 expression in pediatric T-ALL samples [133]. Furthermore, EPHB6-expressing cells were shown to be significantly selected in minimal residual disease up to 30 days from the standard treatments, while they also presented increased markers related to cell proliferation and poor clinical outcome, such as cyclin B1 (CCNB1) and kinesin family member 20A (KIF20A) [133]. For the case of pediatric B-cell ALL (B-ALL), Kong et al. identified *EFNB1* among the downregulated differentially expressed genes (DEGs) enriched in the cell cycle process [134]. El-Sisi et al. investigated the expression of *EPHA4* by RT-PCR in peripheral blood samples of a cohort that included 58 acute myeloid leukemia (AML) patients, among whom 19 were children, demonstrating a positive expression in 36.8% of pediatric AML patients [135]. Another study of bone marrow mononuclear cells from pediatric AML patients reported decreased EPHB1 phosphorylation and mRNA expression compared to the healthy controls, with the majority of AML specimens presenting hypermethylation of the *EPHB1* promoter. Ephrin-B1 ligation to EPHB1 induced p53 DNA binding, which restored the DNA damage response (DDR) via activation of ataxia-telangiectasia mutated (ATM)- and Rad3-related (ATR), Chk1, p53, p21, p38, cyclin-dependent kinase 1 (CDK1) (tyr15), and Bax, and the downregulation of heat shock protein (HSP)27 and Bcl2. When *EPHB1* expression was reintroduced in *EPHB1*-methylated AML cells, the same cascade of ATR, Chk1, p21, and CDK1(tyr15) was upregulated, promoting programmed cell death. It can, therefore, be proposed that in pediatric AML, EPHB1 exerts its tumor-protective role via its effect on DDR [136]. The anticipated involvement of EPHs/ephrins in pediatric hematologic malignancies, given their roles in hematopoietic differentiation, necessitates further studies to validate and further explore the aforementioned findings. The reported implications of EPHs/ephrins in hematopoietic cell development and neoplastic transformation are summarized in Table 5. 

## 5. Discussion 

The involvement of EPHs and their membrane-bound ligands, ephrins, in the physiologic development and homeostasis of human tissues partially explains their involvement in all stages of tumorigenesis, as the dysregulation of developmental and homeostatic mechanisms (angiogenesis, cell-cell interactions, migration, and proliferation) may result in abnormal cell activity as a natural consequence [26,31,33,34,35,38,39]. Thus far, research has mainly focused on the role of the EPH/ephrin signaling pathway in multiple malignancies affecting the adult population, unraveling both tumor-promoting and tumor-protective features and highlighting correlations with disease characteristics, response to treatment, and prognosis [44,50,137,138,139,140,141,142]. In this review, we aim to investigate the implication of the EPH/ephrin axis in pediatric neoplasms, which has been barely investigated compared to its study in older populations.

Variable patterns of EPH/ephrin expression have been identified in cells and tissues from different pediatric malignancies, either solid tumors or hematologic neoplasms. Certain EPH/ephrin members present high expression levels in the setting of neoplasia compared to normal tissues. EPHA2 shows overexpression predominantly in pediatric CNS tumors and malignancies arising from bone and soft tissues, namely glioma [73,74,75], medulloblastoma cell lines [67], craniopharyngioma [97], *NF2*-deficient meningioma [98], Ewing sarcoma [114], and osteosarcoma [113,114]. A subset of pediatric AML cases is characterized by EPHA4 expression in peripheral blood samples [135]. EPHB1 and EPHB3 are overexpressed in rhabdomyosarcoma tumor cells [121], while EPHB2 high-level expression has been reported in medulloblastoma tumor tissue samples [66,67], in *NF2*-deficient meningioma cell lines [98], and rhabdomyosarcoma tumor cells [121]. Increased EPHB3 levels have also been reported in ependymoma cells [95]. Neuroblastoma, medulloblastoma, and rhabdomyosarcoma are characterized by increased EPHB4 expression [106,121], while T-ALL presents enhanced EPHB6 expression [133]. Analogous findings have been reported for ephrins, with increased expression of ephrin-A3 and ephrin-A4 in ependymoma cell lines [95], of ephrin-B1 in medulloblastoma cell lines [67] and rhabdomyosarcoma tumor cells [120], and of ephrin-B2 in gliomas [80] and rhabdomyosarcoma tumor cells [120]. Comparable in number are the reports on the tumor-suppressive properties of the EPH/ephrin axis in pediatric neoplasms. Specifically, the observation that a high expression of EPHA2 in neuroblastoma [103], EPHB1 in glioma [79], EPHB6, ephrin-B2, and ephrin-B3 in neuroblastoma [101] correlates with a better prognosis is in line with a tumor-suppressive role of these EPHs. *EFNB1* has been identified among downregulated DEGs in children diagnosed with AML [134], while bone marrow samples from pediatric AML patients are characterized by *EPHB1* promoter hypermethylation with subsequent decreased *EPHB1* expression [136], while analogously for the case of pediatric ALL, the observed increased *EPHB4* methylation resulting in decreased EPHB4 expression may contribute to leukemia development and progression [131]. Interestingly, as it is observed for the case of EPHB4, the same EPH can exert either a pro-tumorigenic (i.e., in neuroblastoma and rhabdomyosarcoma [106,120]) or anti-tumorigenic function (i.e., in ALL [131]) depending on the type of neoplasm it is expressed. We should note, though, that for certain EPH/ephrin members, such as EPHB1 in medulloblastoma [69] and EPHB2 in ependymoma [95], the reported variable levels of expression render the characterization of their role in tumorigenesis ambiguous. It is also worth notable that variations or dysregulations prevalent in pediatric oncology has not currently been established; however, some members are dysregulated at a high frequency in many types of solid neoplasms, specifically EPHA2, EPHB1, EPHB2, EPHB3, ephrin-B2, and ephrin-B3 [42,67,69,71,73,74,75,82,83,95,97,98,101,102,103,114,120,121]. 

Of note, the expression pattern of EPHs and ephrins can prove useful in the diagnostic evaluation and better characterization of specific subtypes of certain pediatric malignancies. More precisely, high levels of EPHA2 and EPHB4 expression could distinguish high-grade gliomas [67,73,91,106] and *NF2*-deficient meningiomas [74]. For the latter, EPHB1 and EPHB2 could also serve as diagnostic markers [75]. As for neuroblastoma, low EPHA2 expression can discriminate the N-type from the S-type, which is characterized by high EPHA2 expression [98], while low EPHB4 levels are typical in embryonal rhabdomyosarcoma [98]. Additionally, the study of EPH/ephrin expression could predict tumor characteristics. Namely, EPHA2 expression in osteosarcoma has been correlated with invasive capacity [103] and increased ephrin-B3 in glioma with more aggressive behavior [122].

Moreover, EPH/ephrin members can serve as prognostic biomarkers in the setting of different pediatric neoplasms, with their expression correlating both with dismal and favorable prognosis, depending on the EPH/ephrin and the tumor subtype. A poor prognosis has been documented for pediatric patients with gliomas overexpressing EPHA7 [86], alveolar-type rhabdomyosarcomas overexpressing EPHB4 [75,81], and osteosarcomas presenting ephrin-A4 cytoplasmic expression [124]. On the other hand, high expression levels of EPHA2 in neuroblastoma [123], of EPHB1 in glioma [120], and of EPHB6, ephrin-B2 and ephrin-B3 in neuroblastoma [103] confer a better prognosis. It is also noteworthy that EPH/ephrin expression could also be used to distinguish patients who are likely to respond to the currently applied treatment modalities. Characteristic examples are the decrease in *EPHA2* expression after doxorubicin treatment in responding patients with neuroblastoma [97], the increased expression of EPHB1 in medulloblastoma radiosensitive tumors [101], the correlation of EPHB6 overexpression in T-ALL with sensitivity to doxorubicin [103], and the association of ephrin-A4 cytoplasmic expression in osteosarcoma and an inferior response to chemotherapy [71]. 

Considering the cardinal role of the EPH/ephrin signaling pathway in a multitude of embryonic developmental processes, such as neurogenesis and neuronal migration [132], angiogenesis and lymphangiogenesis [120], as well as morphogenesis and cell differentiation in various tissues [33], its dysregulation can significantly contribute to the development of pediatric malignancies affecting multiple tissues. The extensive research on EPHs and ephrins in pediatric tumors of the nervous system probably stems from their thoroughly established function in neuronal differentiation, migration, and proliferation [1,38,60,61,62,64]. Nevertheless, the contribution of the EPH/ephrin axis in homeostatic functions of postnatal life highlights that aberrations may favor tumorigenesis in practically any tissue and throughout childhood. Despite the extensive investigation of EPH/ephrin expression patterns in pediatric tumors, studies addressing specific interactions between EPHs and ephrins in pediatric tumors are few. Namely, in medulloblastoma, ephrin-B1 has been shown to induce the phosphorylation of EPHB2 and to a lesser extent, EPHB4 [70]; EPHB4/ephrin-B2 signaling in glioma has been identified to promote angiogenesis [67,71], while Ephrin-A1-induced phosphorylation of EPHA2 has been reported to participate in osteosarcoma tumorigenesis [90]. Increased expression or activation of EPHs can induce downstream signaling cascades, such as those mediated by AKT, ERK, MAPK, mTOR, Rac1, and Cdc42 [67,70,71,74,91,113] in CNS tumors, PKCγ, ERK1-2, and CDK4, with c-RAF and VEGF as target genes in neuroendocrine tumors [76,77], and Akt and MAPK in sarcomas [104,105,106], that overall enhance tumor cell proliferation and invasive properties. Of note, as proven in the case of pediatric AML, the EPH/ephrin signaling pathway is implicated in DDR [112], with aberrations probably enhancing the mutational burden.

Eph/ephrin members’ variety of expression and involvement in tumorigenesis, which are distinct in some tumor types, make them potential targets of therapeutic interventions. Antibodies targeting members of the EPH/ephrin system have already been tested in clinical trials [113]. Interestingly, many drugs that are used off-label for the treatment of different cancer subtypes in the pediatric population target EPH/ephrin members: Panobinostat, used in brain tumors and sarcomas targeting EPHB1, EPHB2, and EPHB3; Dasatinib, used in brain tumors targeting EPHB1, EPHB2 [87,112,136], and in neuroblastoma targeting EPHB4 [97]; and Pazopanib and Trametinib targeting EPHA2 in rhabdomyosarcoma [98]. We should note that the mechanism of action of the above-mentioned drugs is not specific for EPH targeting, and the observed effects on EPHs are probably a result of universal histone deacetylation for the case of Panobinostat and multi-targeted tyrosine kinase inhibition for the case of Dasatinib, Pazopanib, and Trametinib. Interestingly, EPH inhibition can be exploited to enhance the effectiveness of the currently applied treatment modalities, considering that EPHB1 knockdown in medulloblastoma cells has been shown to enhance cellular radiosensitization [106]. To our knowledge, there are no clinical trials on pediatric populations specifically targeting the EPH/ephrin signaling pathway. However, as mentioned above, targeted agents are used off-label in mainly solid pediatric neoplasms (e.g., Dasatinib, Trametinib), and, moreover, novel therapies are used in the fields of clinical trials (e.g., Regorafenib in Ewing sarcoma) and multi-target tyrosine kinase (and non-tyrosine kinase) receptors, including EPH/ephrin members [143,144,145]. Other therapeutic strategies that need to be tested in pediatric populations after presenting promising results in older patients are antibody–drug conjugates (e.g., anti-EphA3 antibody–drug conjugate, which has shown promising results in glioblastoma multiforme, EphB2 antibody-2H9, which has shown promising results in colorectal cancer) and small molecules targeting protein–protein interactions, specifically the polyphenol and steroidal derivatives, α1 agonist doxazosin, 2,5-dimethylpyrrol- 1-yl-benzoic acids, and amino acid conjugates of lithocholic acid, which are potential tumor suppressors (e.g., resulting in EPHA2-mediated tumor suppression in prostate cancer cells) [146,147,148,149].

Overall, the EPH/ephrin system is involved in a variety of pediatric neoplasms, offering several potential approaches for developing novel prognostic markers and Eph-targeted tumor-selective therapies. These approaches are favored by EPH/ephrin members’ more studied and proven roles in adults’ malignancies. As it is described for the pediatric population, in adults, the EPH/ephrin signaling pathway has tumor-promoting (e.g., EPHB2 in breast tumors) or tumor-suppressing functions (e.g., EPHB6 in lung tumors) and is involved in a variety of tumors (e.g., hepatocellular carcinoma, colorectal cancer, melanoma, non-small cell lung carcinoma, and thymic tumors) [138,139,140,141,142,147]. However, in adult studies, tumor samples were extensively screened, and several mutations were associated with cancer development (e.g., p35-36 region of chromosome 1 for EPHB2) [150]. It is finally worth mentioning that, in adults, a small number of EPHs (specifically EPHA2, EPHA3, EPHA4, and EPHB4) are considered promising therapeutic targets, similarly with pediatric patients; however, a greater variety of therapeutic options has been tested in adult populations as it was previously described herein (antibody–drug conjugates, peptides, and small-molecule inhibitors) [147,151,152,153]. Further studies are required to clarify the complex mechanisms underlying the EPH/ephrin axis implication in the pathogenesis of pediatric neoplasms, with this better understanding being irrefutably necessary for the development of novel therapeutic approaches. 

## 6. Limitations

The present study has several limitations. First of all, we have included some studies referring to neoplasms with a high frequency in childhood (specifically glioma and osteosarcoma), in which the study material is retrieved from adult populations. Evidence from studies in pediatric patients regarding the EPH/ephrin system’s involvement in these types of neoplasms is limited, and we aimed to shed light on the currently available evidence as an impetus for further investigation, especially regarding targeting EPHA2 with Dasatinib, Pazopanib, and Trametinib, which have been already used off-label in pediatric solid tumors with promising results [72,74,75,76,77,78,79,80,116,117,118,119,120]. Moreover, pharmaceutical targeted agents referred to herein (Dasatinib, Regorafenib, Trametinib, Pazopanib, and Panobinostat) target not only the EPH/ephrin members but a wide panel of receptor tyrosine kinases, e.g., EGFR, VEGFR, discoidin domain receptor 1 (DDR1), mitogen-activated protein kinase (MEK) and non-receptor tyrosine kinases, e.g., Fyn-related kinase (FRK), Breast tumor kinase (BRK), and the positive results are possibly a combination of multiple-receptor targeting [154,155,156,157,158]. Another limitation of our study is that the evidence presented is mainly observed from preclinical studies in cell lines, human tissue material, and mouse models; clinical data from large patient cohorts and pediatric-oriented well-designed experimental models, which are not currently available, are needed to confirm the results presented herein. Finally, we do not draw conclusions regarding the role of the EPH/ephrin system in the mutational profile of pediatric neoplastic subtypes, as the exact percentages of mutations in EPH/ephrin members across the age spectrum of pediatric and adulthood populations with neoplastic disease have not been established yet and is a potential field for further research via large genomic analyses. 

## 7. Conclusions—Future Perspectives

The study of the pathogenetic mechanisms underlying tumorigenesis and metastasis in pediatric neoplasms represents an evolving research field. The detection of new molecular pathways involved in these processes can reveal novel diagnostic and prognostic biomarkers, as well as potential therapeutic targets. Towards this direction, identifying the expression patterns of EPHs and ephrins in pediatric neoplasms and deciphering their downstream signaling cascades, which can either promote or inhibit tumor development and progression, can contribute to the emergence of treatment modalities to address unmet needs, especially in tumor subtypes that currently remain intractable with the standard therapeutic options.

## Figures and Tables

**Figure 1 ijms-25-03834-f001:**
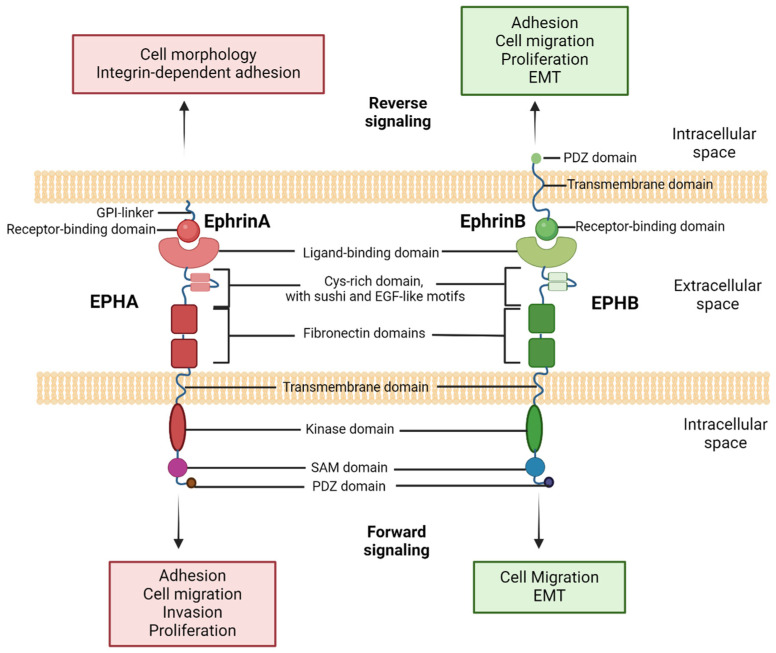
Structure and interactions of EPHs and ephrins. EPHs bind membrane-bound ephrins; therefore, cell–cell contact is generally necessary for receptor activation. Type A ephrins bind to EPHAs, and type B ephrins bind to EPHBs, though exceptions to this general rule have been documented. EGF, epidermal growth factor; EMT, epithelial–mesenchymal transition; GPI, glycosylphosphatidylinositol; PDZ, post-synaptic density protein (PSD95), Drosophila disc large tumor suppressor (Dlg1) and zonula occludens-1 protein; SAM, sterile alpha motif. Created with BioRender.com.

**Figure 2 ijms-25-03834-f002:**
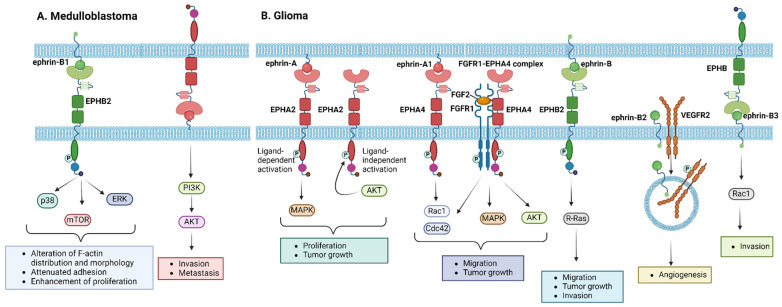
Molecular pathways induced by EPH/ephrin members in the pathogenesis of (**A**) medulloblastoma and (**B**) glioma. Cdc42, cell division control protein 42 homolog; ERK, extracellular-signal-regulated kinase; FGF2, fibroblast growth factor 2; FGFR1, fibroblast growth factor receptor 1; MAPK, mitogen-activated protein kinase; mTOR, mammalian target of rapamycin; PI3K, phosphoinositide 3-kinase; VEGFR2, vascular endothelial growth factor receptor 2. Created with BioRender.com.

**Figure 3 ijms-25-03834-f003:**
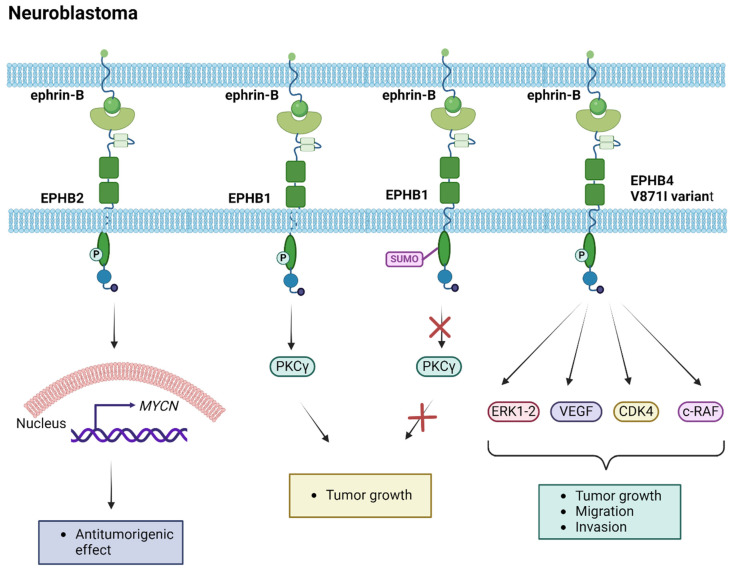
Molecular pathways induced by EPH/ephrin members in the pathogenesis of neuroblastoma. CDK4, cyclin-dependent kinase 4; ERK, extracellular-signal-regulated kinase; PKCγ, protein kinase C gamma; SUMO, small ubiquitin-like modifier; VEGF, vascular endothelial growth factor. Created with BioRender.com.

**Figure 4 ijms-25-03834-f004:**
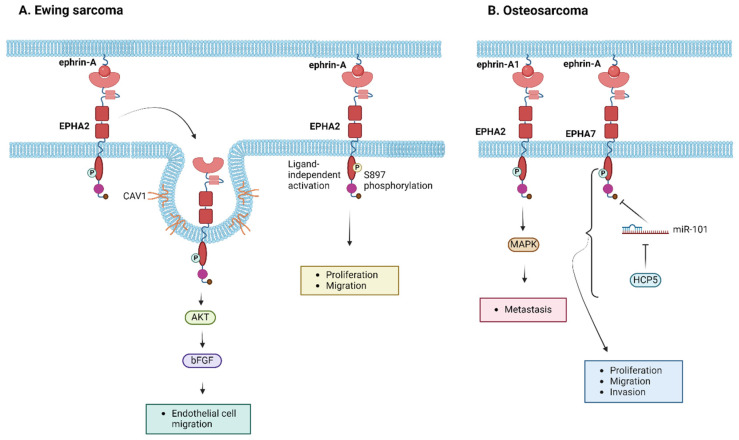
Molecular pathways induced by EPH/ephrin members in the pathogenesis of (**A**) Ewing sarcoma and (**B**) osteosarcoma. bFGF, basic fibroblast growth factor; CAV1, caveolin-1; HCP5, histocompatibility leukocyte antigen complex P5; MAPK, mitogen-activated protein kinase; miR, microRNA. Created with BioRender.com.

**Table 1 ijms-25-03834-t001:** The role of EPH and ephrin family members in the central nervous system normal development and tumor pathogenesis, and their potential therapeutic targeting.

EPH/Ephrin Member	Embryonic Developmental Function	Postnatal/Adult Normal (Homeostatic) Function	Tumor Type	Expression Pattern in Tumor	Tumorigenic Function	Potential Targets and Targeting Effect
EPHA2		CNS angiogenesis inhibition via interaction with ephrin-A1 [60]	Glioma	High-level expression in high-grade gliomas [73,74,75]	Promotion of neoplastic cell migration and proliferation [73,74,85] Ligand-mediated EPHA2 activation acts via MAPK-dependent pathway [74] Ligand-independent EPHA2 activation via phosphorylation at serine 897 by Akt [76]	Targeting with Dasatinib results in downregulation of proliferation in mouse and human cell lines [87]
			Medulloblastoma	High-level expression in medulloblastoma cell lines compared to normal cells [67]		
			Craniopharyngioma	High-level expression compared to normal cells [97]		Targeting with Dasatinib and Regorafenib has shown promising effects in tumor inhibition [97]
			*NF2*-deficient meningioma	Overexpression in *NF2*-deficient meningioma cell lines compared to normal cells [98]		Targeting with Dasatinib results in a dose-dependent reduction in expression [98]
			Diffuse intrinsic pontine glioma			Targeting with *Panobinostat results in a dose-dependent reduction in cell invasion [92]*
EPHA4	Motility enhancement of migrating medial ganglionic eminence inter-neurons. [54]	1. Self-renewal of radial glial cells in the neocortex via interaction with ephrin-B1 [60] 2. Postnatal and adult neural stem cell preservation in an undifferentiated state in vivo via interaction with ephrin-A5 [61] 3. Varying levels of expression in normal cerebellum cells [70]	Glioma		Inhibition of neoplastic cell migration and proliferation [85] EPHA4 promotes FGF2-mediated cell proliferation and migration and increased MAPK and Akt phosphorylation stimulated by FGF2 [77] EPHA4 overexpressing cells show increased active forms of Rac1 and Cdc42 [77] EPHA4 forms a heteroreceptor complex with FGFR1 inducing FGFR1-mediated downstream signaling [77]	
			Diffuse intrinsic pontine glioma			Targeting with *Panobinostat results in dose dose-dependent reduction in cell invasion [92]*
EPHA7		1. Neuronal migration and proliferation in subventricular zone (SVZ) [64] 2. Multiple-level expression in normal cerebellum cells [70]	Glioma	High-level expression in poor-prognosis gliomas [75]	*Enhancement of tumorigenic effects [75]*	
EPHB1	1. Retinal ganglion cell axons direction to the ipsilateral trajectory [57] 2. Dendritic spine formation and synaptic maturation [57] 3. Neuron migration enhancement [59]	1. Neuron antiproliferative effect [60] 2. Neuron proliferation control in SVZ [64]	Glioma	High-level expression in favorable prognosis gliomas [67]	*Inhibition of neoplastic cell migration and invasion [85]*	
			Medulloblastoma	Varying levels of expression in 95% of the tumors [69] High-level expression in radiosensitive tumors [71]	*Tumorigenic activity via enhancement of cell proliferation, cell migration, and upregulation of survival proteins [70]*	
			Diffuse intrinsic pontine glioma			Targeting with *Panobinostat results in a dose-dependent reduction in cell invasion [92]*
			*NF2*-deficient meningioma	Overexpression in all *NF2*-deficient meningioma cell lines [98]		Targeting with Dasatinib results in a dose-dependent reduction in expression [98]
EPHB2	Dendritic spine formation and synaptic maturation [57]	Early migration of subgranular zone (SGZ) progenitors [60]	Medulloblastoma	High-level expression in tumor tissue samples [66,67]		
			Ependymoma	Varying levels of expression in 95% of the tumors [42]	*Expression required for ependymoma development [96]*	
			*NF2*-deficient meningioma	Overexpression in *NF2*-deficient meningioma cell lines [98]		Targeting with Dasatinib results in a dose-dependent reduction in expression [98]
			Diffuse intrinsic pontine glioma			Targeting with *Panobinostat results in a dose-dependent reduction in cell invasion [92]*
EPHB3	Dendritic spine formation and synaptic maturation [57]	Antiproliferative effect [60]	Ependymoma	High-level expression in ependymoma cells [95]		
			Diffuse intrinsic pontine glioma			Targeting with *Panobinostat results in a dose-dependent reduction in cell invasion [92]*
EPHB4			Medulloblastoma	High-level expression in most medulloblastoma cell lines [67]		
			High-grade glioma	High-level expression [91]		*Cell proliferation and invasion* downregulation via EPHB4 stimulation with an ephrin-B2 fusion protein [91]
ephrin-A2		Negative regulator of adult neurogenesis [62]	Glioma		*Expression associated with a favorable prognosis [72]*	
ephrin-A3		Negative regulator of adult neurogenesis [62]	Ependymoma	*High-level expression in ependymoma cell lines [95]*		
ephrin-A4			Ependymoma	*High-level expression in ependymoma cell lines [95]*		
ephrin-A5		1. Postnatal and adult neural stem preservation cells in an undifferentiated state in vivo via interaction with EPHA4 [61] 2. Hippocampal neurogenesis and vascular formation in SGZ [63]	Medulloblastoma		*Expression associated with tumor growth and invasive capacity [70]*	
ephrin-B1	Inhibition of apical progenitors’ adhesion via GTPase Arf6 negative regulation [57]	Self-renewal of radial glial cells in the neocortex via interaction with EPHA4 [60]	Medulloblastoma	*High-level expression in medulloblastoma cell lines [67]*	*Association with high invasive capacity via induction of EPHB2 phosphorylation and p38, ERK, and mTOR downstream signaling [66,67]*	
ephrin-B2			Medulloblastoma		Expression associated with high invasive capacity [68,69]	
			Glioma	High-level expression [82]	Tumor invasion and angiogenesis via VEGFR2 regulation [79,81]	
ephrin-B3			Glioma	High-level expression in aggressive glioma [81]	Tumor invasion via Rac1 activation [79,81]	

Cdc42, cell division control protein 42 homolog; CNS, central nervous system; FGF2, fibroblast growth factor 2; FGFR1, fibroblast growth factor receptor 1; NF2, neurofibromatosis type 2; SGZ, subgranular zone; SVZ, subventricular zone; VEGFR2, vascular endothelial growth factor receptor 2.

**Table 2 ijms-25-03834-t002:** The role of EPH and ephrin family members in peripheral nervous system normal development and tumor pathogenesis, and their potential therapeutic targeting.

EPH/Ephrin Member	Embryonic Developmental Function	Postnatal/Adult Normal (Homeostatic) Function	Tumor Type	Expression Pattern in Tumor	Tumorigenic Function	Potential Targets and Targeting Effect
EPHA2			Neuroblastoma	Low-level expression in N-type neuroblastoma cell lines [103]		Targeting with Doxorubicin results in increased expression in neuroblastoma cells [103]
				High-level expression in S-type neuroblastoma cell lines [103]		
				High-level expression in neuroblastoma cells associated with better prognosis [103]		
EPHB1			Neuroblastoma		*Tumorigenesis via PKCγ activation [105]*	
EPHB2		Schwann cells guidance in collective migration, forming multicellular cords that facilitate axon guidance across the injury site [89]	Neuroblastoma	Expression in low-stage tumors correlated with *MYCN* expression [102]		
EPHB4			Neuroblastoma	Expression in neuroblastoma cells [106]	EPHB4-V871I variant associated with increased proliferation, migration, and invasion properties via the ERK1-2 pathway and VEGF, c-RAF, and CDK4 target genes [106]	Pharmaceutical targeting inhibits kinase activity obstructing tumor progression in neuroblastoma cell lines [106]
EPHB6			Neuroblastoma	Expression in tumor specimens associated with a favorable prognosis [101]	*Tumorigenesis via involvement in axon guidance pathways [104]*	
ephrin-B2			Neuroblastoma	Expression in tumor specimens associated with a favorable prognosis [101]		
ephrin-B3	Neurons migration [59]		Neuroblastoma	Expression in tumor specimens associated with a favorable prognosis [101]		

CDK4, cyclin-dependent kinase 4; ERK, extracellular-signal-regulated kinase; PKCγ, protein kinase C gamma; VEGF, vascular endothelial growth factor.

**Table 3 ijms-25-03834-t003:** The role of EPH and ephrin family members in bone and soft tissue normal development and tumor pathogenesis, and their potential therapeutic targeting.

Eph/Ephrin Member	Embryonic Developmental Function	Postnatal/Adult Normal (Homeostatic) Function	Tumor Type	Expression Pattern in Tumor	Tumorigenic Function	Potential Targets and Targeting Effect
EPHA2			Ewing sarcoma	Overexpression in Ewing sarcoma compared to normal tissue, especially in males [114]	1. EPHA2 and CAV1 interaction induces AKT-mediated bFGF production, *promoting endothelial cell migration angiogenetic activity* [112] 2. EPHA2 phosphorylation correlated with the capacity of Ewing sarcoma cell migration [86]	
			Osteosarcoma	High-level expression in osteosarcoma, especially in males and in tumors with high invasive capacity [113,114]	Ephrin-A1 binding to EPHA2 induces the activation of MAPK-mediated downstream signaling pathways, stimulating osteosarcoma’s metastatic potential [113]	Pazopanib, Trametinib [115,116,117]
EPHA7			Osteosarcoma		miR-101 binding to EPHA7 regulates EPHA7 activation—HCP5 competitive binding to miR-101 inhibits EPHA7-induced cell proliferation, migration, and invasion [118]	
EPHB1			Rhabdomyosarcoma	Expression in rhabdomyosarcoma tumor cells [121]		
EPHB2		Mesenchymal stem cell migration [107]	Rhabdomyosarcoma	Expression in rhabdomyosarcoma tumor cells [121]		
EPHB3	Bone growth in the embryo (especially at the calvaria suture fronts, periosteum, chondrocytes, and trabeculae of developing long bones) [107]		Rhabdomyosarcoma	Expression in rhabdomyosarcoma tumor cells [121]		
EPHB4		Angiogenesis stimulation [107]	Rhabdomyosarcoma	Expression in rhabdomyosarcoma tumor cells [121]		
				High-level expression in alveolar type associated with poor prognosis [123,124]		
				Low-level expression in embryonal rhabdomyosarcoma [122]		
ephrin-A4			Osteosarcoma	Cytoplasmic expression in osteosarcoma cells associated with poor prognosis and inferior response to chemotherapy [120]		
ephrin-B1		Bone remodeling [107]	Rhabdomyosarcoma	Expressed in rhabdomyosarcoma tumor cells [120]		
ephrin-B2	VEGF-angiogenesis control during development [109,110]	1. Bone remodeling [97] 2. Expressed by osteoclasts [109]	Rhabdomyosarcoma	Expressed in rhabdomyosarcoma tumor cells [120]		

bFGF, basic fibroblast growth factor; CAV1, caveolin-1; HCP5, human histocompatibility leukocyte antigen complex P5; MAPK, mitogen-activated protein kinase.

**Table 4 ijms-25-03834-t004:** The role of EPH and ephrin family members in renal normal development and tumor pathogenesis, and their potential therapeutic targeting.

Eph/Ephrin Member	Embryonic Developmental Function	Postnatal/Adult Normal (Homeostatic) Function	Tumor Type	Expression Pattern in Tumor	Tumorigenic Function	Potential Targets and Targeting Effect
EPHB2	Involved in genesis, differentiation, and developmental process of renal system structures (glomerular epithelium, cloaca, and urethra) [125]	Venous and capillary maintenance and angiogenic remodeling [127]	Wilms Tumor	Increased expression in tumor samples of advanced tumor stages [127]		

**Table 5 ijms-25-03834-t005:** The role of EPH and ephrin family members in normal and malignant hematopoiesis and their potential therapeutic targeting.

EPH/Ephrin Member	Embryonic Developmental Function	Postnatal/Adult Normal (Homeostatic) Function	Tumor Type	Expression Pattern in Neoplastic cells	Tumorigenic Function	Potential Targets and Targeting Effect
EPHA4		Lymphocyte development [129]	AML	Positive expression in 36.8% of peripheral blood samples of pediatric AML cases [134]	Tumor-promoting [134]	
EPHB1		1. HSPC maintenance 2. HSPC differentiation 3. Erythroid differentiation [129]	AML	Decreased EPHB1 phosphorylation and mRNA expression with concurrent *EPHB1* promoter hypermethylation in AML bone marrow samples compared to healthy controls [136]	Tumor-suppressive-EPHB1 activation induces DDR [136]	
EPHB4	Fetal B-cell maturation	1. Myeloid Differentiation 2. Erythroid differentiation 3. Adult B-cell maturation 4. Monocyte differentiation 5. Adhesion, migration, and extravasation of monocytes and macrophages [122]	ALL	Higher EPHB4 methylation in ALL samples compared to controls [131]	Tumor-suppressive [131]	
EPHB6		Thymic T-cell development [129]	T-ALL	High-level expression associated with sensitivity to chemotherapy [132]	EPHB6-mediated preservation of Akt signaling increases chemosensitivity [132]	Doxorubicin [132]
				High-level expression [133]	Chemotherapy resistance via enhancement of self-proliferation processes (increased CCNB1 and KIF20A) [133]	
ephrin-B1		1. HSPC maintenance 2. B-cell development 3. T-cell development [129]	Β-ALL	Among downregulated DEGs enriched in the cell cycle process [132]	Tumor-suppressive [132]	

AML, acute myeloid leukemia; B-ALL, B-cell acute lymphoblastic leukemia; CCNB1, cyclin B1; DEGs, differentially expressed genes; DDR, DNA damage response; HSPC, hematopoietic stem/progenitor cell; KIF20A, kinesin family member 20A; T-ALL, T-cell acute lymphoblastic leukemia.

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
