# Peer review of "The Clinical Relevance of the EPH/Ephrin Signaling Pathway in Pediatric Solid and Hematologic Malignancies"

_ijms, 2024, doi:10.3390/ijms25073834_

Round 1
Reviewer 1 Report
Comments and Suggestions for Authors
In the manuscript titled "The Clinical Relevance of the EPH/ephrin Signaling Pathway in Pediatric Solid and Hematologic Malignancies", the authors discuss the role of the EPH/ephrin Signaling Pathway in various types of cancer. While the topic is interesting, there are a few issues that need to be considered.
Firstly, the authors focus on pediatric malignancies, but the evidence for the role of EPH/ephrin Signaling in pediatric malignancies is limited. Most of the available evidence is derived from studies conducted on cancer in adults or at the cellular level, with only a few references directly providing evidence for the role of EPH/ephrin Signaling in pediatric malignancies.
Secondly, the authors should discuss the evidence for different types of mutations in EPH/ephrin Signaling receptors in different types of pediatric cancers. Is there any data available to show what percentage of children with different types of malignancies carry a mutation in EPH/ephrin Signaling receptors?
Lastly, different strategies that can be employed to target EPH/ephrin Signaling in cancer treatment should be discussed in more detail.
Author Response
Dear Reviewer
Thank you for taking the time to review our manuscript and for providing constructive criticism. Your insights are invaluable to us and will undoubtedly enhance the quality and relevance of our work.
We are pleased to inform you that we have carefully addressed each of the points you raised in your review. Specifically, we have:
Reviewer Comment 1: Firstly, the authors focus on pediatric malignancies, but the evidence for the role of EPH/ephrin Signaling in pediatric malignancies is limited. Most of the available evidence is derived from studies conducted on cancer in adults or at the cellular level, with only a few references directly providing evidence for the role of EPH/ephrin Signaling in pediatric malignancies.
Response: Since the evidence for some common types of pediatric malignancies is limited, we have
included some studies in adult and young adult populations, for the reasons provided in the
“Limitations” paragraph (lines 684-705) that has been added, lines 685-692.
Reviewer Comment 2: Secondly, the authors should discuss the evidence for different types of mutations in EPH/ephrin Signaling receptors in different types of pediatric cancers. Is there any data available to show what percentage of children with different types of malignancies carry a mutation in EPH/ephrin Signaling receptors?
Response: To our knowledge, there is no solid evidence regarding the exact percentage of mutations in Eph/ephrin in pediatric population, as referred in lines 700-705.
Reviewer Comment 3: Lastly, different strategies that can be employed to target EPH/ephrin Signaling in cancer treatment should be discussed in more detail.
Response: Regarding your comment about discussing strategies for targeting the EPH/ephrin Signaling pathway in cancer treatment in more detail, we would like to inform you that we have addressed this concern by incorporating a dedicated paragraph on this topic. You can find this paragraph between lines 651-664, where we elaborate on various strategies for targeting the Eph/ephrin pathway. We believe that this addition enhances the comprehensiveness of our manuscript. Once again, we extend our gratitude for your thoughtful review and constructive feedback, which have undoubtedly improved the overall quality of our manuscript.
Reviewer 2 Report
Comments and Suggestions for Authors
The manuscript (Manuscript ID:ijms-2914743) entitled “The Clinical Relevance of the EPH/ephrin Signaling Pathway in Pediatric Solid and Hematologic Malignancies” revolves around pediatric neoplasms (PN), which present a series of unique challenges in diagnosis and treatment. Overall, the manuscript is written well and the figures are properly articulated to represent the essence of the scientific concept underlying the personalized diagnostic and therapeutic strategies hypothesized. However additional modifications/ suggestions might improve the standard of the manuscript further.
- Are there any known variations or dysregulations within the EPH/ephrin pathway that are particularly prevalent or significant in pediatric oncology?
- How do the findings regarding the EPH/ephrin pathway in pediatric neoplasms compare with its role in adult malignancies?
- Are there any experimental models or systems that were used to study the impact or role of the EPH/ephrin cascade in PN, and how accurately do they recapitulate the clinical scenario?
- Considering the bidirectional nature of the EPH/ephrin axis, are there any potential therapeutic strategies targeting this pathway that have shown promise in preclinical or clinical studies for pediatric cancer treatment?
- What are the limitations of the study? How can this concept be translated into practical research? If so, What could be the potential limiting factors?
Moderate editing of English language required
Author Response
Dear Reviewer
Thank you for taking the time to review our manuscript and for providing constructive criticism. Your insights are invaluable to us and will undoubtedly enhance the quality and relevance of our work.
We are pleased to inform you that we have carefully addressed each of the points you raised in your review. Specifically, we have:
Reviewer Comment 1: Are there any known variations or dysregulations within the EPH/ephrin pathway that are particularly prevalent or significant in pediatric oncology?
Response: Thank you for your inquiry. While our study didn't uncover specific Eph/ephrin variations prevalent in pediatric oncology, it documented the involvement of certain Eph/ephrin members in various pediatric neoplasms. We have now incorporated these findings into our manuscript, as suggested, and they are addressed in more detail in lines 581-585.
Reviewer Comment 2: How do the findings regarding the EPH/ephrin pathway in pediatric neoplasms compare with its role in adult malignancies?
Response: Thank you for your insightful question regarding the comparison of findings regarding the EPH/ephrin pathway in pediatric neoplasms versus its role in adult malignancies. In response to your query, we have included a dedicated paragraph that compares the involvement of Eph/ephrin in pediatric and adult cancers. This addition can be found between lines 666-683 of our manuscript.
Reviewer Comment 3: Are there any experimental models or systems that were used to study the impact or role of the EPH/ephrin cascade in PN, and how accurately do they recapitulate the clinical scenario?
Response: Thank you for your question regarding the use of experimental models or systems to study the impact or role of the EPH/ephrin cascade in pediatric neoplasms (PN), and their accuracy in recapitulating the clinical scenario. We would like to clarify that, as referenced in lines 697-700 of our manuscript, no experimental models have been specifically established for studying Eph/ephrin signaling in pediatric malignancies.
Reviewer Comment 4: Considering the bidirectional nature of the EPH/ephrin axis, are there any potential therapeutic strategies targeting this pathway that have shown promise in preclinical or clinical studies for pediatric cancer treatment?
Response: Thank you for your thoughtful comment. We have addressed your inquiry by incorporating relevant information in the document. Specifically, we have added text directing attention to lines 637-664, where the available evidence on potential therapeutic strategies targeting the EPH/ephrin axis in pediatric cancer treatment, along with promising results, is detailed. We hope this addition satisfactorily addresses your question. If you have any further inquiries or require additional clarification, please don't hesitate to let us know.
Reviewer Comment 5: What are the limitations of the study? How can this concept be translated into practical research? If so, What could be the potential limiting factors?
Response: Thank you for your insightful comment regarding the limitations of our study and the potential for practical research translation. We have taken your feedback into consideration and have made appropriate revisions to the document. Specifically, we have included a dedicated paragraph highlighting the limitations of our study, which can be found in lines 684-705. Should you have any further questions or require additional clarification, please feel free to reach out.
Reviewer 3 Report
Comments and Suggestions for Authors
The authors did a very good job in writing a comprehensive review about EPH and ephrins with a focus on pediatric malignancies. The flow of the review is solid, the tables and the figures are relevant and informative. The review covers general aspects and goes deeper in each pediatric cancer, contextualizing the various actors of the EPH/ephrin at play, their effects, also highlighting where the effects of the signaling could be opposite.
The overall judgment of this article is positive, and there is not much to point out to improve it. Some of the minor points that come to the mind are that in some places, especially towards the first pages of the article, some words or sentences could be improved. As an example, see line 57, 80 (simultaneously), 89, 105.
At line 119 where the authors mention that there could be crosstalk with other signaling systems, it would be nice to have some examples to give context to the reader, and also give a "dimension" of the phenomena, if it is as common and important as the normal signaling, or the contrary.
Also, it could be beneficial if the article pointed out more explicitly information about single-cell studies where EPH and ephrin were observed or researched. This would add value to the reviewing in pointing the readers towards datasets of interest, and also summarizing them.
These are details that could improve an already good article.
Comments on the Quality of English LanguageAs mentioned above, some sentences and word could benefit from being reviewed to make the text sound more natural and fluent in English.
Author Response
Dear Reviewer
Thank you for taking the time to review our manuscript and for providing constructive criticism. Your insights are invaluable to us and will undoubtedly enhance the quality and relevance of our work.
We are pleased to inform you that we have carefully addressed each of the points you raised in your review. Specifically, we have:
Reviewer Comment 1: The overall judgment of this article is positive, and there is not much to point out to improve it. Some of the minor points that come to the mind are that in some places, especially towards the first pages of the article, some words or sentences could be improved. As an example, see line 57, 80 (simultaneously), 89, 105.
Response: We improved our writing in lines 57,80,89,105 of the previous article, as asked, and
moreover, in other lines towards the first pages, as asked (lines 50,71,107-110,114,546-547)
Reviewer Comment 2: At line 119 where the authors mention that there could be crosstalk with other signaling systems, it would be nice to have some examples to give context to the reader, and also give a "dimension" of the phenomena, if it is as common and important as the normal signaling, or the contrary.
Response: We shortly analyzed the crosstalk between Eph/ephrin system and other communication systems the way asked, lines 122-130.
Reviewer Comment 3: Also, it could be beneficial if the article pointed out more explicitly information about single-cell studies where EPH and ephrin were observed or researched. This would add value to the reviewing in pointing the readers towards datasets of interest, and also summarizing them.
Response: We pointed recent single-cell analysis examples summarizing their findings, lines 172-180.